Swabs to genomes: a comprehensive workflow

Dunitz Madison I. 1
Lang Jenna M. 1
Jospin Guillaume 1
Darling Aaron E. 2
Eisen Jonathan A. 1 jaeisen@ucdavis.edu
Coil David A. 1
1 UC Davis, Genome Center , USA
2 ithree institute, University of Technology Sydney , Australia
Kumar Abhishek
Electronic publication date: 2015 May 14
Publication date: 2015
Volume: 3
Electronic Location ID: e960
Received 2015 Jan 30; Accepted 2015 Apr 24
Copyright: © 2015 Dunitz et al.
Copyright year: 2015
Copyright holder: Dunitz et al.
License: This is an open access article distributed under the terms of the Creative Commons Attribution License, which permits unrestricted use, distribution, reproduction and adaptation in any medium and for any purpose provided that it is properly attributed. For attribution, the original author(s), title, publication source (PeerJ) and either DOI or URL of the article must be cited.
License URL: https://creativecommons.org/licenses/by/4.0/

Keywords: Workflow, Microbial genomics, Genome sequencing, Genome assembly, Bioinformatics

Funding: Alfred P. Sloan Foundation This work was funded by the Alfred P. Sloan Foundation as part of their “Microbiology of the Built Environment” program. The funders had no role in study design, data collection and analysis, decision to publish, or preparation of the manuscript.

==============================
The sequencing, assembly, and basic analysis of microbial genomes, once a painstaking and expensive undertaking, has become much easier for research labs with access to standard molecular biology and computational tools. However, there are a confusing variety of options available for DNA library preparation and sequencing, and inexperience with bioinformatics can pose a significant barrier to entry for many who may be interested in microbial genomics. The objective of the present study was to design, test, troubleshoot, and publish a simple, comprehensive workflow from the collection of an environmental sample (a swab) to a published microbial genome; empowering even a lab or classroom with limited resources and bioinformatics experience to perform it.

Introduction

Thanks to decreases in cost and difficulty, sequencing the genome of a microorganism is becoming a relatively common activity in many research and educational institutions. However, such microbial genome sequencing is still far from routine or simple. The objective of this work was to design, test, troubleshoot, and publish a comprehensive workflow for microbial genome sequencing, encompassing everything from culturing new organisms to depositing sequence data; enabling even a lab with limited resources and bioinformatics experience to perform it.

In late 2011, our lab began a project with the goal of having undergraduate students generate genome sequences for microorganisms isolated from the “Built Environment”. The project focused on the Built Environment because it was part of the larger “microBEnet” (microbiology of the Built Environment network, www.microbe.net) effort. This project serves many purposes, including (1) engaging undergraduates in research on microbiology of the Built Environment, (2) generating “reference genomes” for microbes that are found in the Built Environment, and (3) providing a resource for educational activities on the microbiology of the Built Environment. As part of this project, undergraduate students isolated and classified microbes, sequenced and assembled their genomes, submitted the genome sequences to databases housed by The National Center for Biotechnology Information (NCBI), and published the genomes (Lo et al., 2013; Bendiks et al., 2013; Flanagan et al., 2013; Diep et al., 2013; Coil et al., 2013; Holland-Moritz et al., 2013). Despite the reduced cost of genome sequencing and the availability of diverse tools making many of the steps easier, (e.g., kits for library prep, cost-effective sequencing, bioinformatics pipelines), there were still a significant number of stumbling blocks. Moreover, some portions of the project involve choosing between a wide variety of options (e.g., choice of assembly program) which can create a barrier for a lab without a bioinformatician. Each option comes with its own advantages and disadvantages in terms of complexity, expense, computing power, time, and experience required. In this workflow, we describe an approach to genome sequencing that allows a researcher to go from a swab to a published paper (Fig. 1). We used this workflow to process a novel Tatumella sp. isolate and publish the genome (Dunitz et al., 2014). The data from every step of the workflow, using this Tatumella isolate, is available on Figshare (Coil, 2014).

Figure 1 Overview of the workflow.

All the steps required to go from a swab to a genome.

The sequencing and de novo assembly of genomes has yielded enormous scientific insight revolutionizing a wide range of fields, from epidemiology to ecology. Our hope is that this workflow will help make this revolution more accessible to all scientists, as well as present educational opportunities for undergraduate researchers and classes.

There are several excellent resources that focus on smaller portions of this entire workflow. Examples include the Computational Genomics Pipeline (Kislyuk et al., 2010) and a “Beginner’s guide to comparative bacterial genome analysis” (Edwards & Holt, 2013). Clarke et al. (2014) describes a similar pipeline focused on human mitochondrial genomes.

Background

Background: bioinformatics

Command line/terminal tutorial

This workflow is written assuming that the user is using a computer running Mac OS X or Linux. It is also possible to carry out many of the computational parts of this workflow in a Windows environment but getting these steps to work in Windows is outside the scope of this project.

Some parts of this workflow require the user to provide text instructions for software programs by using a command line interface. While potentially intimidating to computer novices, the use of command line interfaces is sometimes necessary (e.g., some programs do not have graphical interfaces) and is also sometimes much more efficient. To access the command line on a Mac open the Terminal program (the default location for this program is in the “Utilities” folder under “Applications”).

When this application is launched, a new window will appear. This is known as a “terminal” or a “terminal window”. In the terminal window, you can interact with your computer without using a mouse. Many popular programs have a GUI (Graphical User Interface) but some programs used in this workflow will not. So, instead of double-clicking to make a program run, you will type a command in the terminal window. Throughout this tutorial, we will instruct you to type commands, but copying and pasting them (when possible) will reduce the occurrence of typos. We will walk you through how to run all of the programs required for this workflow, but you must first acquire a basic familiarity with how to interact with your computer through the terminal window. Below is a list of commands that will be required to use this workflow. There are many tutorials available to help you get started.

For more information on operating in the terminal, check out this informative video: https://www.youtube.com/watch?v=zRZT4nQP3sE.

And this interactive tutorial: http://www.ee.surrey.ac.uk/Teaching/Unix/.

Summary of Unix/Linux commands and terms

$ ls lists files and directories (folders). If left as just “ls” this command will list the files and directories in your current location. If a “path” is added afterwards (e.g., ls /usr) this command will list the files and directories in that location.

$ cd use to change directories

$ cd .. use to move up one directory

$ cd directory_name use to move to that directory

$ cd ∼ use to move to the home directory of the current user

$ grep “some pattern” file_name displays lines that match the pattern (contained within the quotes) for which you are searching. If a line contains the same character multiple times it will only be displayed once

$ grep –c “what you want to count” file_name counts the number of lines containing a specific character or sequence of characters

$ less file_name view a file, type q to exit

A few quick definitions:

command line—the command line is where you type commands in a terminal window

script—a computer program. Usually computer programs are called scripts when they perform relatively simple functions that are limited in scope. Scripts are typically only run from the command line

directory—a folder

compile—turning a human-readable file into a computer-executable program

Software updates

Software packages are updated with varying frequencies. Some such updates will render the instructions offered here obsolete. When this occurs, you should consult with the software manual for help. An internet search with a description of the problem you are having may prove helpful. Another option is to email the software developer; many are remarkably responsive. As a last resort, consult with a colleague who is more comfortable with bioinformatics or computer programming. Most software updates will require only minor modifications. For example, we might provide you with instructions to type:

./software_1.2.0/software.py

but a more recent release might necessitate:

./software_1.3.0/software.py

Background: molecular biology and microbiology

This workflow assumes a basic knowledge of molecular biology and sterile technique (methods for carrying out lab experiments without contamination from living microorganisms). The starting point is the collection of microbes from a surface with a swab. We will cover the steps necessary to take a sample through plating, dilution streaking, overnight growth, creating a glycerol stock, 16S rDNA PCR, and preparation for Sanger sequencing to determine the identity of your bacterial or archaeal isolate.

Throughout the ‘Isolation’ section we refer frequently to “media” and “culture media”. This is in reference to the type of substrate (sometimes liquid, sometimes a gel-like material such as agar) used to grow microbes in the lab. The choice of media will depend on the goals of the particular project. Some factors to consider when selecting media and conditions for growth include:

1. What type of organism do you want to isolate?

2. Are there types of organisms (e.g., pathogens) that you would prefer not to isolate? For example, swabbing people and growing samples on blood agar at 37 °C can preferentially isolate human pathogens.

3. How much time is available for growth and isolation?

• growth rates differ both between organisms (e.g., species 1 versus species 2) and also in different conditions for the same organisms (e.g., species 1 at 20 °C vs. 37 °C)

• for many microbes there is an “optimal growth temperature” (OGT—the temperature at which it grows best) but the OGT varies between species

• you will be able to isolate a greater diversity of organisms if you allow a long time for slow-growing organisms to grow

4. What types of equipment are available to you?

• if an organism grows most happily at 37 °C, then you will need to have an incubator and shaker available at that temperature.

For our previous work we used a rich media, lysogeny broth (LB), and growth at either room temperature (∼25 °C) or 37 °C. For some basic information on media preparation and agar plates, we recommend the following resource: http://teach.genetics.utah.edu/content/gsl/html/agar.html.

Background: phylogeny and systematics

In order to identify to which organism a 16S rDNA sequence belongs, as well as to provide an evolutionary context for your organism of interest, we recommend inferring a phylogenetic tree (see ‘Building a 16S rDNA Tree’). Building such a phylogenetic tree is (relatively speaking) the easy part. Intelligent interpretation of the tree will require an investment of time, similar to the investment required to learn the basics of UNIX. Fortunately, there are a number of resources available for this purpose. We recommend this online tutorial (http://evolution.berkeley.edu/evolibrary/article/phylogenetics_02) or Baldauf (2003). Here we provide a brief introduction to phylogenetic trees.

A phylogenetic tree is a diagram representing a model of evolutionary relationships. Phylogenetic trees have three main components: taxa, branches, and nodes (Fig. 2). These are defined below:

Figure 2 A model phylogenetic tree.

A phylogenetic tree is often helpful in assigning taxonomy to an unknown sequence.

• Taxon. An individual or grouping of individuals. This could be individual sequences, species, families, phyla, etc. For phylogenetic analyses, the taxa that are drawn at the tips of branches are sometimes referred to as “leaves” on the tree.

• Branch. A representation of the evolution of a taxon over time (sometimes also known as an evolutionary lineage). There are three main types of branches in a tree. Terminal branches are those that lead to the tips or leaves in the tree. Internal branches connect branches to each other. And the root branch, also known as the root of the tree, is the branch that leads from the base of the tree to the first node in the tree.

• Node. These are the points where individual branches end. In the internal parts of a phylogenetic tree, single branches can “split” producing multiple descendant branches. The point at which the branches split is known as an internal node. If a branch ends at a taxon, the end point is known as a “terminal node”.

Some other information to know about trees:

• Clade. A group of organisms consisting of a single node and all the descendants of that node in a tree and nothing else.

• Bootstrapping. A statistical method used to measure how well a node is supported by all the data being used.

• Ingroup. The group of taxa being studied.

• Outgroup. A taxon that separated in an evolutionary tree prior to the existence of the most recent common ancestor of the ingroup.

Isolation

This section will take you through the basics of isolating, culturing, and storing your organism.

Swab

Using a sterile cotton swab (for example the “Sterile Cotton Tipped Applicators” from Puritan), wipe (i.e., “swab”) the area you intend to sample for 10–15 s, as if you were trying to clean the area. Try to rotate the swab to ensure that all sides touch the surface.

Plate

Gently (so as not to break the agar surface) rub, i.e., “streak” the swab across the entire surface of an agar plate. Be sure to rotate the swab as you are doing so to ensure that all sides of the swab make contact with the plate. Incubate the plate at the desired temperature (in our case, usually 37 °C or room temperature) until colonies appear.

Dilution streak (streaking for individual colonies) x2

After incubation, choose desired colonies (we typically attempt to maximize the diversity of colony morphologies) and dilution streak them onto individual plates. Dilution streaking involves spreading out a chosen colony such that new single colonies grow on a new plate (details can be found online).

After growth to visible colonies, repeat the dilution streaking to help ensure purity of the culture. Some organisms will only grow in tight association with others, and a mixed culture will prove difficult to classify and assemble.

Liquid culture

After the second dilution streaking, a liquid culture is needed both for long-term storage and for DNA extraction. Transfer a single colony from each dilution streak plate into 5 mls of culture media and grow for 1–3 days until cloudy. Once the liquid culture is ready, prepare a 10% final concentration glycerol stock for long-term storage at −80 °C from 1 to 2 ml of the sample.

16S rDNA Sanger Sequencing

Following liquid culturing, the organisms need to be identified, or classified. This is accomplished by determining and then analyzing the DNA sequence of the 16S rRNA gene. In this section, we describe how the sequence of this gene is determined and readied for analysis. The general outline is as follows: DNA extraction, polymerase chain reaction (PCR) amplification of the 16S rRNA gene, and sequencing of the resulting PCR product using Sanger sequencing (Sanger, Nicklen & Coulson, 1977). There are multiple approaches one can take to these steps. For example, the PCR requires DNA from the organism of interest. That DNA can come directly from a liquid culture of the organism (when this is used for PCR this is known as colony PCR). Alternatively, one can take a liquid culture and then isolate the DNA from that culture and use the purified DNA as input material for the PCR. This adds an extra step to the process—a step known as DNA extraction (see below). Colony PCR significantly decreases the amount of work needed for preparation, but it can yield poorer results, both in terms of PCR success and resultant sequence quality. However, we recommend colony PCR when screening a large number of samples. DNA extraction can then be used for any recalcitrant samples. DNA extraction is significantly more work, but it often generates better Sanger sequences allowing for more accurate identification.

DNA extraction

There are a number of different options for DNA extraction, and which one should be used depends on many factors including available equipment, experience, and cost. A standard approach in microbiology involves the use of a phenol and chloroform extraction followed by ethanol precipitation, and any number of protocols for this approach can be found in books, articles and on the internet. A common alternative approach is to use a commercially available kit—there are many advantages to such kits—notably ease and lack of toxic chemicals. A disadvantage of kits is that they typically are more expensive per sample than other approaches (especially if one is only doing a few samples, since most kits include materials for a minimum of 50 samples). For most projects, we use kits—typically the Promega-Wizard Genomic DNA Purification Kit.

Follow the protocol or kit instructions provided by the manufacturer and then proceed to “PCR” below.

Colony PCR (if not extracting DNA)

Centrifuge 1 ml of the overnight culture until the cells form a pellet at the bottom of the tube (about 5 min at 10,000 g), pour off the liquid on top (the supernatant) and resuspend the pellet in 100 µl of sterile DNAase-free water. Incubate the samples at 100 °C for 10 min to help lyse the cells. Use the resulting solution as the template in the PCR below.

PCR

This reaction uses the 27F (AGAGTTTGATCMTGGCTCAG) and 1391R (GACGGGCGGTGTGTRCA) primers which amplify a near full-length bacterial (and many archaeal) 16S rRNA gene. Our lab uses standard PCR reagents (Qiagen or Kappa), with an annealing temperature of 54 °C and an extension at 72 °C of 90 s. Do not forget to include positive (any sample containing bacterial genomic DNA that you have successfully amplified before) and negative (e.g., replace DNA with water) controls. The full program we use is:

1: 95 °C for 2:00

2: 95 °C for 0:15

3: 54 °C for 0:30

4: 72 °C for 1:30

5: Go to 2 (40 times)

6: 72 °C for 3:00

7: 4 °C forever.

After PCR is completed, confirm the PCR worked by agarose gel electrophoresis, all controls behaved as expected (i.e., band in the positive control and no band in the negative control), and that you have DNA fragments of the correct size (∼1,350bp).

Submit samples for sequencing

Very few single-researcher labs currently have the capacity to do Sanger sequencing. However, there are a number of DNA sequencing facilities (commercial and academic) that provide Sanger sequencing services for researchers. They will handle as little as a single sample, or will allow you to submit an unlimited number of samples, arrayed in 96-well plates. You will typically provide both your PCR product as well as primers for sequencing (the same primers used for PCR are usually used for sequencing). To get the most data, do not forget to request both forward (e.g., using primer 27F) and reverse (e.g., using primer 1391R) reactions for each sample. Each facility will have its own guidelines concerning DNA and primer concentration. Our lab uses the UC Davis Sequencing Facility. If an internet search does not reveal the presence of a Sequencing Facility near you, most sequencing centers will allow you to ship samples to them for sequencing. Another possibility is Science Exchange which is an online clearinghouse for lab services.

Sanger Sequence Processing

The end product of Sanger sequencing is the production of sequences (reads) for each sample submitted. Upon receiving Sanger reads from a sequencing facility, typically as .abi files via email, it is necessary to do some pre-processing before they can be analyzed. These steps include quality trimming the reads, reverse complementing the reverse sequence, aligning the reads, generating a consensus sequence, and converting to FASTA format. There are very limited options for free software that allow the user to perform these steps.

In this workflow we recommend using an automated pipeline available at the Ribosomal Database Project (Cole et al., 2013) if working with a large number of sequences. This pipeline only provides a rough view, since it doesn’t orient or align the reads, it simply quality trims them and outputs the data in a format that can be fed directly to the BLAST program at NCBI (Altschul, 1990). This will at least give an idea of which genera, and sometimes which species, to which each sequence can be classified. We then recommend processing samples of interest using SeqTrace (Stucky, 2012) which allows the user to see the traces (graphical representation of reads), process the sequences manually, and a get a longer, more accurate sequence for analysis.

We have also created a script that will perform the same steps as SeqTrace automatically, but does not allow you to adjust any of the parameters. The choice of our script (easy, little control) versus SeqTrace (more complex, more control) will depend on the user and the project.

RDP sanger pipeline

(Recommended as a starting place, or when working with many sequences)

The RDP Sanger analysis pipeline can be found here https://rdp.cme.msu.edu/login/pipeline/libSummary.

This pipeline allows you to upload one zipped folder containing multiple .abi traces. It cleans and processes the sequences and generates a FASTA file of the processed sequences; which can then be uploaded to BLAST and analyzed. This allows you to quickly screen your samples before running the files through the more time consuming SeqTrace analysis which will reverse complement and align the reads to generate a consensus sequence.

After signing in to RDP, you will be on the “Library Run Summary” page. Click on the “Create New Run” tab near the top of the page. Select the appropriate 16S rRNA gene (Archaea or Bacteria depending on your sample) name your library and choose a library name abbreviation and select any vector (this pipeline assumes cloned PCR fragments but will work fine regardless of what you select here). Select the “Upload the data without well mapping” button at the bottom of the page. You will now be directed to the “Data Loader” page, choose a zipped folder containing the .abi traces you wish to analyze and click “Load Data”. To create the folder, put all of the .abi traces you are working with into a folder, right click on the folder and select Compress “folder name”. If you downloaded the files as a group from your sequencing facility, they may already be in a zipped folder.

When the pipeline is finished, you will be directed to click a link that will open a new window containing the library run stats. Select the “Download Raw Sequence” button. Navigate to http://blast.ncbi.nlm.nih.gov/Blast.cgi?PROGRAM=blastn&PAGE_TYPE=BlastSearch&LINK_LOC=blasthome and select the “Choose File” button underneath the area for the FASTA sequence. Select the file you just downloaded from the library run stats page. We recommend checking the box to exclude Uncultured/environmental sample sequences then click “BLAST”. If you are working with a large number of FASTA sequences, it may take a few minutes. When the BLAST search is complete, you can cycle through the results using the pull down menu to the right of the “Results for:” heading.

SeqTrace

We recommend using SeqTrace first if only working with a couple of sequences. When working with a large batch it might be easier to do a preliminary screening of the sequences using the RDP Sanger pipeline above and only using SeqTrace for sequences of interest.

Download the program from https://code.google.com/p/seqtrace/downloads/list.

Installation directions https://code.google.com/p/seqtrace/wiki/Installation.

Installing and running SeqTrace on a PC is simple; installing it on a Mac requires a few more steps than for a PC. The installation guide offers two options for installing SeqTrace on a Mac; we recommend running SeqTrace with native GTK+.

To install SeqTrace on a Mac, you will need to download the PyGTK package from OSX. http://sourceforge.net/projects/macpkg/files/PyGTK/2.24.0/PyGTK.pkg/download.

Currently, SeqTrace depends on Python version 2.x. Confirm that you have Python version 2.x. You can do this by typing:

python --version

You should see something that looks like “Python 2.6.9” If you see Python 3.x, seek outside help to run an earlier version. http://www.python.org/downloads/.

After downloading and unpacking the program, SeqTrace is ready for use. SeqTrace must be launched from a Terminal window. For a refresher or introduction to the Terminal, see ‘Background: bioinformatics’. Move SeqTrace to your Applications folder.

Open a terminal window and copy/paste or type:

/Applications/seqtrace-0.9.0/seqtrace.py

This syntax will only work if the SeqTrace folder’s name is seqtrace-0.9.0, if you saved it under a different name you will need to replace seqtrace-0.9.0 with the name of that folder.

This will launch SeqTrace from the terminal in a Python shell; you will need to keep the terminal window open while you are using the program.

SeqTrace provides excellent instructions for using the program at https://code.google.com/p/seqtrace/wiki/WorkingWithProjects.

Edit and create a consensus sequence with seqTrace

For this workflow we have found that the following is the simplest way to edit and create a consensus sequence from forward and reverse reads in SeqTrace.

1. Create a new project (File > New Project). Add your forward and reverse primer sequences here; we used 27F (AGAGTTTGATCMTGGCTCAG) and 1391R (GACGGGCGGTGTGTRCA) and click “OK”.

2. To add files, go to “Traces” and click on “Add trace files”, then select the reads (.abi files) you want to work with.

3. The program is able to recognize forward and reverse reads from information in the file name if they are properly formatted.

• Go to “Traces” and click on “Find” and mark forward/reverse. The default setting looks for _F for forward and _R for reverse. This can be edited in the Project settings (you can pull it up by clicking on the picture of the tool at the top of the page) and changing the search strings under trace settings. For an example, see Fig. 3.

• If the program is able to recognize the forward/reverse reads it will place an orange left pointing arrow in front of reverse reads and a blue right pointing arrow in front of forward reads. This step is not necessary to get a consensus sequence, it just makes organizing the reads easier.

Figure 3 SeqTrace options.

This screenshot shows an example of manually entered primer information in SeqTrace.

1. Pull up the “Project Settings” by clicking on the picture of tool at the top of the page. Click on the “Sequence Processing” tab and under “Sequence trimming”, unclick the Automatically trim sequence ends button. You should also decrease the Min. confidence score under “Consensus” settings. The default option is 30, which represents 99.9% accuracy. For many reads this will be too stringent and will not allow you to get enough overlap to create a consensus sequence. A minimum confidence score between 15 and 25 is normally okay but tuning may be required depending on your read quality. For an example, see Fig. 4.

2. Group your forward and reverse reads by highlighting both of them and clicking “Group selected forward/reverse files” (under “Traces”).

3. Under “Sequences” go to “Generate Finished Sequences” and click on “for all trace files”. (You will need to redo this every time you change the project settings.)

4. To view your consensus sequence, click on the read pair group and then click on the magnifying glass at the top of the page. You should see something like Fig. 5.

5. The “Trace View” shows the quality scores, the chromatogram (trace) display, and the raw base calls from both the forward and reverse reads, as well as the consensus sequence. The consensus sequence is the middle list of nucleotides. If the program is giving you a string of Ns where your forward and reverse reads do not overlap, you need to decrease the Min. confidence score.

6. To export the consensus from the trace view, go to “Sequence”, hover on “Export Sequences”, and select “Export Sequences from Selected Trace Files”. This will create a file containing the consensus sequence, which can then be used for analysis such as searching for closely related sequences using the BLAST program (Altschul, 1990) which can be used to identify the organism.

Figure 4 SeqTrace trimming setting.

An example of reducing the minimum confidence score in SeqTrace.

Figure 5 Sanger chromatogram.

This screenshot from SeqTrace shows both the chromatogram (trace) as well as the consensus sequence.

Custom script to create a consensus sequence (merge_sanger_16s.pl)

This custom script is for users who prefer to quickly trim and align their sequences. It is to be used in place of SeqTrace, with or without having pre-screened samples using the RDP Sanger pipeline described above.

Download/install

1. Create a new folder called “Sanger_seq” on your Desktop.

2. Download the zip file, containing three scripts (merge_sanger_16s.pl, cleanup.pl and subsample_reads.pl) from Coil, Jospin & Lang (2014).

3. Open the zip file and move the merge_sanger_16s.pl file to the new “Sanger_seq” folder.

MUSCLE

In order to run this script you will need to download MUSCLE (Edgar, 2004) from here: http://www.drive5.com/muscle/downloads.htm. Uncompress and open the MUSCLE directory, and record the full path and name of the executable file (e.g., muscle3.8.31_i86darwin64) for later use.

Convert files from .abi to .fastq

To run the merge_sanger_16s.pl script, you will first need to convert your read files from .abi to .fastq.

This can be done at http://sequenceconversion.bugaco.com/converter/biology/sequences/.

Use the drop down menus to set it to convert .abi files to .fastq. Upload a file and convert it. The converted file will save to your downloads folder under the name “sample.fastq”. If you are working with a lot of reads, we recommend immediately renaming the files to match the original .abi file name to avoid confusion.

Edit and create a consensus sequence

Once all of your files are in .fastq format, move all of them to the “Sanger_seq’ folder in which you saved the merge_sanger_16s.pl script. Use the terminal to navigate to within this folder by typing:

cd /Desktop/Sanger_seq

Then, to run the script, type:

perl merge_sanger_16s.pl <muscle_path> <file1.fastq> <file2.fastq>

muscle_path refers to the full path and file name that were recorded earlier (e.g., /Users/username/Downloads/muscle_download/muscle/muscle3.8.31_i86darwin64)

The script will return one of 2 messages:

1. “Found N conflicting case(s) during merging of X residues”

2. “Not enough data to overlap confidently”

In the first case, the merging happened, however there may be some conflicting bases. The fewer the better. It can be an indication of how confident the user should be with the results. Since this is a very crude method, it should be noted that there is no complex algorithm behind the merge. There is a simple comparison for which we keep the base that had the highest quality score.

In the second outcome, the sequences were trimmed too much when doing the quality-trimming. The length of both sequences end to end was smaller than the fragment length that we are looking for. This is an indication of poor quality sequence and most users should not proceed (others can lower the quality threshold set by the script).

The newly merged file will be saved as file1_merged.fasta and can be uploaded to BLAST for identification (see ‘BLAST 16S rDNA sequence’).

Organism Identification Using 16S rRNA Gene Sequence

It is necessary to screen the 16S rDNA Sanger sequencing results for possible genome sequencing candidates. We recommend starting with BLAST results, then continuing onto the Genomes Online Database (GOLD). This is a large database containing most sequenced genomes and many ongoing sequencing projects. Sometimes the use of GOLD and an internet search will be sufficient to obtain information about the organism you have isolated. In many cases, it will be useful to build a phylogenetic tree to aid in identification, as the BLAST search results may not be sufficiently informative.

BLAST 16S rDNA sequence

Begin by navigating to the Standard Nucleotide BLAST at NCBI: http://blast.ncbi.nlm.nih.gov/Blast.cgi?PROGRAM=blastn&PAGE_TYPE=BlastSearch&LINK_LOC=blasthome.

Paste in your Sanger consensus sequence. We recommend checking the box to exclude Uncultured/environmental sample sequences, since these will not be informative for identification. Be sure the nucleotide collection (nr/nt) is selected under database and click the “BLAST” button (Fig. 6).

Figure 6 BLAST options.

The recommended settings for using BLAST in this workflow.

Interpreting the results

Depending on the quality of the Sanger sequencing and the particular microbe sequenced, the BLAST search results can range from definitive to relatively uninformative. Examples of both are discussed below.

1. In some cases, it is not necessary to build a phylogenetic tree for further identification. If all of the top hits are the same species (or end in sp.), have e-values of 0.0, good query coverage, and 99%–100% identity, you can proceed to “Using GOLD”.

2. In other cases, the results are more ambiguous. The results may show more than 99% identity to multiple species, or even to multiple genera. In this case, refer to ‘Building a 16S rDNA Tree’, before using GOLD.

3. Another possibility is that you will get significantly less than 99% identity to any sequences in the NCBI database. One explanation for this is that your sequence is of poor quality. This might require more stringent trimming using SeqTrace or even resequencing if the quality is poor enough to make assigning taxonomy difficult. Another possibility is that you have isolated something that is not very closely related to anything in the NCBI database. In the latter case, we would recommend first re-doing the BLAST search, but unchecking “Uncultured/environmental sample” to see if the sequence matches others that have been found, but are not associated with a cultured organism. In either case, we would recommend re-sequencing for confirmation and then refer to ‘Building a 16S rDNA Tree’ to examine the phylogenetic context of the novel sequence.

Using GOLD (the genomes online database)

Go to: http://genomesonline.org/cgi-bin/GOLD/index.cgi.

Under the Search tab, click the “Quick Search” option and you should be taken to a page that looks like the screen shot displayed in Fig. 7.

Figure 7 GOLD search.

Sample “Quick Search” page on GOLD.

Fill out the Biosample name section, with information about your microbe from BLAST and click “Sequencing Project Search”. We usually search for only the genus to get a sense for how well that genus is represented in the database and which species are present. Figure 8 shows an example screen shot of the results for “Brachybacterium”. The third column (Project Status) lists the current status of the project (complete, permanent draft, incomplete, targeted). While some “incomplete” and “targeted” projects will be completed, many will not, so we tend to ignore these categories.

Figure 8 GOLD results.

Sample results for Brachybacterium on GOLD.

If you have relatively ambiguous identification results (e.g., you think you have some sort of Brachybacterium but aren’t sure which species,) it could be worthwhile to perform an alignment of your 16S rDNA sequence with those from genomes already in Genbank or to build a phylogenetic tree as in ‘Building a 16S rDNA Tree’.

Compare two 16S rDNA sequences

First locate the 16S rRNA gene sequences of the genome you’d like to compare to, by searching the NCBI Nucleotide database using the name of your species and “16S ribosomal RNA”.

http://www.ncbi.nlm.nih.gov/nuccore/.

Click on the sequence of interest, then click on the “FASTA” link to get the sequence in FASTA format. Now navigate to the “Align Sequences Nucleotide BLAST” page: http://blast.ncbi.nlm.nih.gov/Blast.cgi?PAGE_TYPE=BlastSearch&BLAST_SPEC=blast2seq&LINK_LOC=align2seq.

Paste in the two 16S rDNA sequences and click on the “BLAST” button. Unless both your sequence and the sequence to which you are comparing were amplified with the same primers, the query coverage will not be 100% (Fig. 9). A low identity can be the result of poor sequence quality or taxonomic distance.

Figure 9 Sample “Align Sequences Nucleotide BLAST” results.

In this example, our sequence of interest is 98% identical to the target sequence.

Choosing an organism to sequence

A choice of whether to sequence an organism based on these results depends on the project goal. How closely your isolate is related to an organism with a sequenced genome might be completely irrelevant if you are interested in sequencing your isolate per se, perhaps because of where you found it or because of some interesting phenotype. If your goal is to increase the phylogenetic diversity of available genome sequences, then sequencing the 200th E. coli genome is not the ideal approach to achieve that goal. At the other extreme, if you have isolated an organism that is only 90% identical to anything with a currently available genome sequence, or that appears to be alone on a long branch on your phylogenetic tree, then you have a good candidate to achieve your goal. Of course 90% is arbitrary, as is a “long” branch, but the current standard is to use a 97% 16S rRNA gene sequence identity as a proxy for species delimitation in bacteria. This is yet another arbitrary cutoff, and is frequently debated in the field (Chan et al., 2012; Drancourt & Raoult, 2005; Hanage, Fraser & Spratt, 2006; Stackebrandt, 2002). Finally, you might be interested in increasing the genome sequences available for a particular lineage, for example, to provide additional data for a comparative genomics project. In that case, and in many others, the ideal number of close relatives and the definition of “close”, will be unique to each project.

Building a 16S rDNA Tree

Our preferred approach to classifying microbial species is to place the unknown organism in the context of a phylogenetic tree using its 16S rRNA gene sequence. Building a phylogenetic tree from a 16S rRNA sequence is fairly straightforward, but the interpretation of the tree can be a bit complex. Here, we attempt to guide you through both. However, some complicated cases will require consultation with an expert in the field of phylogenetics or systematics.

The outline of the workflow is to use the Ribosomal Database Project (RDP) to generate an alignment of the sequence with close relatives and an outgroup. This is followed by cleanup of the RDP headers, tree-building with FastTree (Price, Dehal & Arkin, 2009), and viewing/interpretation of the tree using Dendroscope (Huson & Scornavacca, 2012).

Obtain an RDP alignment

The goal of this section is to obtain an alignment of 16S rRNA gene sequences from RDP that can be used to build a tree. This procedure has the added benefit of providing an independent verification of the taxonomic assignment of your sequence based on the BLAST results.

1. Go to http://rdp.cme.msu.edu.

2. Create an account.

3. Click on “my RDP/login”.

4. Upload the fasta file containing your 16S rDNA sequence.

5. Assign it a group name (this is what the program will label your sequence/organism). Choose this carefully since that will be the name on the final tree.

6. Click the “+” next to the sequence to add it to your cart.

7. Click on “CLASSIFIER” at the top of the page.

8. Click on “Do Classification With Selected Sequences” button. This will show you a hierarchical view of the classification of your sequence (from Phylum to Genus). You will use this information to navigate to other sequences that you want to include in the alignment that you will use to build your phylogenetic tree.

9. Click on “BROWSERS”. We recommend opening “BROWSERS” in a new tab so that you can keep the hierarchy information handy.

10. Click on “Isolates” to select only isolates for further analysis. Then click “Browse” (Fig. 10).

11. Click on the + sign next to “Archaea outgroup”. This will add an Archaeal sequence to your cart, which will be used to root your phylogenetic tree. Even better would be to chose an outgroup within the same bacterial phyla that you know to be outside of the clade you are examining. If in doubt, just use the Archaeal one.

12. If using the example sequence provided, click on “Proteobacteria”, then “Gammaproteobacteria”, then “Enterobacteriales”, then “Enterobacteriaceae”. This will take you to the Genus Tatumella, which currently has over 69 entries in it. If the genus you are working with has too many sequences to analyze easily (for example, Bacillus currently has >26,000), one way to reduce this number is to exclude the uncultured taxa in the database. To do this, scroll down to the “Data Set Options” and click on the “Isolates” button. Click “Refresh” and you will see that there are fewer sequences in the Genus. To reduce this number further, click on the “Type” Strain button (though if you do this you’ll have to build a tree later for species identification since each species will only be represented once in the tree). As a worst-case scenario, you will need to manually select a subset of organisms to include in your alignment.

13. Click on the + sign next to genus Tatumella to add all of those sequences to your cart.

14. Click on “Sequence Cart” and confirm that your uploaded sequence, the outgroup sequence, and all of the other sequences you’d like to include in your tree are displayed.

15. Click on “download”, leave the download options as the defaults (fasta, aligned, uncorrected), and then click on the appropriate download button. Save the file and then rename it to something informative.

Figure 10 RDP options.

Here we show our recommended options for RDP.

Clean up the RDP taxon names

The RDP alignment will have taxon names that most of the downstream software tools will not tolerate because they include special text characters. So, we have written a little Perl script (cleanup.pl) that will remove those special characters and replace them with underscores. This script is included in the zip file of scripts on Figshare (Coil, Jospin & Lang, 2014). To run cleanup.pl, first move it to your Applications folder. Then, in a Terminal window, navigate to the directory that contains the RDP alignment that you’ve just downloaded. Then, type or copy/paste:

perl /Applications/cleanup.pl -i RDP_alignment.fa -o RDP_alignment_clean.fa

Building the tree with FastTree

There are two ways to get FastTree, which will be required for building the tree from your alignment. The first, recommended approach, is to use Phylosift (installed below in ‘Assessing completeness with Phylosift’) which contains a working version of FastTree. In this case, you will simply call the program from the Phylosift directory with the following command (be sure the path to Phylosift calls the correct version):

/phylosift/osx/FastTree -nt RDP_alignment_clean.fa > tree_file.tre

The other option is to install FastTree directly, which is a bit more involved.

Go to http://www.microbesonline.org/fasttree/#Install and download the FastTree.c program by right clicking on it and saving the link to your Applications folder. To compile the software, navigate to your Applications folder in a Terminal window:

cd /Applications

Then, type or copy/paste:

gcc -O3 -finline-functions -funroll-loops -Wall -o FastTree FastTree.c -lm

This compiling of FastTree requires a software tool called gcc (http://gcc.gnu.org). If your attempt to compile FastTree with the instructions above fails, the most likely reason is that you do not have gcc. You can download and install gcc from Xcode here https://developer.apple.com/downloads/index.action?q=xcode.

In order to download Xcode, you will need to register as a developer with Apple which takes only a couple of minutes. After you register, click on the apple next to “Developer” at the top of the page. Then, click on the Xcode download link, which will ultimately take you to the Mac App Store, where you can follow the instructions to install Xcode. Once it is installed, open the program and open preferences (under the Xcode tab). Click on the downloads option and install the command line tools.

Once you have successfully downloaded and installed Xcode and the command line tools, return to your Applications folder in a Terminal window and type or copy/paste again:

gcc -O3 -finline-functions -funroll-loops -Wall -o FastTree FastTree.c -lm

Now, you should have a working version of FastTree. To build your tree, using the cleaned up RDP alignment, type or copy/paste the following (be sure the output name ends in “.tre” to ensure it will be recognized by Dendroscope):

/Applications/FastTree -nt RDP_alignment_clean.fa > tree_file.tre

Viewing the tree in dendroscope

Download and install Dendroscope. http://ab.inf.uni-tuebingen.de/software/dendroscope/.

You will need to obtain a license here http://www-ab2.informatik.uni-tuebingen.de/software/dendroscope/register/.

Enter the license number into Dendrosope and then you can open your phylogenetic tree from the File menu to view it.

Once the tree is visible, the first step is to re-root the tree to the outgroup. Expand the tree by clicking the expansion button (labeled in Fig. 11), then scroll through the tree to locate the outgroup. Click on the beginning of the taxon name, to select it, and reroot the tree by going to edit and selecting “re-root”.

Figure 11 Dendroscope options.

The circle shows the options for expanding/shrinking the tree, while the arrow points to the “phylogram” option.

We recommend viewing the tree as a phylogram, which can be accomplished by clicking on the phylogram button (labeled in Fig. 11). From this tree it should be possible to determine the phylogenetic placement of the candidate sequence, and in some cases to give it a name with more certainty than a simple BLAST search. Below are examples of a relatively informative tree and a relatively uninformative tree:

Figure 12 An informative phylogenetic tree.

This phylogenetic tree shows our sequence of interest to be in a clade where everything has the same name.

In the tree shown in Fig. 12 (genus Brachybacterium), our sample of interest from an assembly is “Brachybacterium muris UCD-AY4” (Lo et al., 2013). It falls within a clade where every named member has the same name “Brachybacterium muris”, and this name does not occur elsewhere on the tree. Hence, we were confident enough to name our organism as that species. In other words, this sequence falls within a well-supported clade of Brachybacterium muris.

In the tree shown in Fig. 13 (genus Tatumella) our species of interest is Tatumella sp. UCD-D suzukii (Dunitz et al., 2014). In contrast to the Brachybacterium example, here our species falls within a poorly defined clade containing multiple species. In this case, we did not assign a species name to this isolate.

Figure 13 An uninformative phylogenetic tree.

In this phylogenetic tree our species of interest is found in a clade with several species, some of which are found in other clades.

Genome Sequencing

Library preparation

The first choice in library preparation is whether to do the library prep yourself or to have the library made by your sequencing provider. The economics of this decision are usually dependent on the number of samples involved. For example, an Illumina TruSeq library prep kit costs around $2,600 for 48 samples. That’s far cheaper than the $150 to $300 that a typical sequencing provider would charge per sample. However, if you’re only preparing a couple of samples there’s no reason to buy an entire kit. The requisite time and ancillary consumables and equipment must also be taken into account (see Table 1). Most sequencing facilities offer library preparation services.

Table 1 Estimated materials costs of bacterial genome sequencing.

This table shows the estimated materials (i.e., without labor) cost of performing a genome sequencing project with this workflow in 2014. The “Best Case” shows the marginal cost of sequencing one genome in a case where you are multiplexing 48 samples, and have the appropriate kits and reagents on hand. The “Worst Case” shows the cost of doing a single genome, with no multiplexing, in a lab where every reagent needed to be purchased new and was not used for anything else.

Projected cost	
Item	Best case (per sample)	Worst case (per sample)	
DNA extractiona	$1.66	$1.66	
PCRb	$0.60	$150	
PCR cleanupc	$2.00	$100	
Sangerd	$14.00	$14	
Library prepe	$58.33	$2,800	
Illumina sequencingf	$35.42	$1,700	
Total	$112.01	$4,930	
Notes.

Specific assumptions are as follows;

a This assumes the purchase of a standard DNA extraction kit, good for 100 samples.

b This assumes purchase of a standard 200U PCR reagent kit.

c PCR cleanup can be performed in a number of ways; gel extraction, beads, or columns for example. Here we assume purchase of a standard column-based kit.

d Sanger sequencing cost is given as the price per reaction ($7 at our sequencing facility), times the forward and reverse reactions.

e This assumes the purchase of a 48-sample Nextera or TrueSeq kit from Illumina, however kits from other manufacturers can be cheaper.

f Our sequencing cost estimate assumes purchase of an Illumina MiSeq run from a sequencing facility.

Kit options

Whether you chose to make libraries yourself or use a service provider, the next major choice is of the type of kit. The two most popular choices with Illumina kits are the Nextera transposase-based kits or the TruSeq kits (with or without PCR). These kits are available from Illumina, but there are also comparable options from other vendors (e.g., New England Biolabs and Kapa Bioscience). In addition, there have been cost-saving modifications to the Illumina kits, for example (Baym et al., 2015). The pros and cons of each type of kit are listed below:

• TruSeq (our recommendation): Pro—The PCR-free protocol minimizes library bias by using mechanical instead of enzymatic DNA fragmentation, and the elimination of PCR results in better assemblies. Con—requires a large amount of DNA (at least 1 µg for PCR-free). There is also now a TruSeq LT kit which only requires 100 ng of DNA and a reduced number of PCR cycles. This may provide a middle option between PCR-free TruSeq and Nextera.

• Nextera: Pro—It allows for very low amounts of input DNA, down to 1ng in the case of the Nextera XT kit. Con—the transposase has an insertion bias and the extensive PCR required for low input samples will also impact the final assembly (Aird et al., 2011).

When growing bacteria in culture as described in this workflow, it should almost always be possible to get enough DNA to use PCR-free TruSeq and therefore minimize library preparation biases in the genome assembly.

Choosing an insert size

In our lab, with paired-end 300bp (PE300) reads on the Illumina MiSeq, we target a DNA fragment size (including adapters) of 600–900bp. The high end of the range is constrained by the maximum length of a DNA molecule that can be amplified on the Illumina MiSeq. The low end of the range is defined by the smallest fragment size that will not produce overlapping reads. Ideally, you would sequence only at the high end of the range because longer insert sizes aid in better genome assembly. However, the range is typically expanded to ensure that enough DNA is available for sequencing. Different sequencing facilities have different opinions on this topic and it is worth having a discussion with your sequencing facility’s point of contact before making any libraries. It is very important that all samples have similar insert sizes if multiplexing as described below.

Multiplexing

Coverage (also known as read depth) is the average number of reads representing a given nucleotide. It is a function of the number and size of genomes pooled onto a run and the number and length of reads. The optimal amount of coverage depends on the read length, the assembler being used, and other factors. The capacity of an Illumina MiSeq with PE300 reads is around 15 Gigabases (Gb), which would result in a coverage of 4300 × for a typical bacterium with a 3.5 Mb genome. On the HiSeq with PE125bp reads, this would be over 14, 000 × coverage. Currently, the recommended coverage for a bacterial genome assembly is 20–200 × depending on the choice of assembler. Therefore, sequencing a single bacterial genome on a full MiSeq or HiSeq run is a significant waste of money and reagents. Furthermore, some current genome assembly algorithms do not perform well given an excess of data, and require down-sampling (i.e., throwing away data, ‘Downsampling’) to achieve the recommended coverage for assembly. We typically multiplex 10–48 genomes on a PE300 MiSeq run and many more on a HiSeq run. If using a kit for library prep, multiplexing is quite straightforward since there are a number of barcoded adapters that come with the kit. We recommend having the sequencing facility demultiplex the samples, as this only requires a list of the barcodes used.

Collaborate

As described above, current Illumina sequencing systems have much greater capacity than is needed for sequencing a single genome. This means it can be generally beneficial to combine many samples into a single run of a machine. Unfortunately, our experience has been that sequencing facilities will typically not help in the coordination of such pooling of samples (we assume because they do not want to oversee the pooling or deal with the associated accounting hassles). Therefore, it is typically up to the users to carry out such coordination. Though this can sometimes be complicated, it is generally worthwhile, since one can pool together many genomes or metagenomes into a single run of a system and still get enough data for each project, thus making the sequencing cost per project significantly lower. For this to work well, one needs to coordinate the use of barcodes to tag each sample, coordinate the pooling, and have available the informatics required to “demultiplex” samples from each other.

Downsampling

For Illumina data assembled using this workflow, we recommend a coverage of between 20 × and 200 ×. See our more detailed discussion in ‘Interpretation of A5-miseq stats”. If you have coverage significantly higher than 200 × and wish to downsample your data, we have written a script (sub_sample_reads) for this purpose. Downsampling should not be necessary if following the assembly instructions in this workflow. If downsampling, you will first need to calculate how many reads you want the script to sample. We recommend determining how many reads would be equivalent to 100 × coverage (divide the genome size by the average read length and multiply by 100). You can download the script from the zipped script file found on Figshare (Coil, Jospin & Lang, 2014). Create a new directory containing the script (sub_sample_reads) and the reads you wish to downsample.

To downsample the data, navigate to the directory you just created (in the terminal) and use the following command

./subsample_reads.pl <file1> <file2> <#_reads_to_keep> <output_file_name>

for example

./subsample_reads.pl test_1.fq test_2.fq 250 my_reads.fastq

For further directions and documentation you can view the script on github.

Genome Assembly and Annotation

Assembly

Genome assembly consists of

1. data pre-processing (quality filtering and adapter removal)

2. error correction

3. contig assembly

4. scaffolding (optional)

5. verification of scaffolds/contigs

The first step simply removes poor quality sequences, as well as adapter sequences left over from sequencing. Some assemblers follow this with error correction where reads are compared to each other to eliminate sequencing errors. Next is contig assembly where overlapping reads are assembled into long continuous stretches of sequences. Scaffolding refers to the alignment and orientation of these contigs relative to each other (where possible). The last step is verification where reads are mapped back to the contigs/scaffolds to reduce misassemblies.

There is a plethora of programs that can perform some, or most of these steps. These programs include commercial and open-source options, some are very user friendly and some are extremely difficult to use/install. Common assemblers for bacterial genomes include SPAdes (Bankevich et al., 2012), MIRA (Chevreux, 2004), SGA (Simpson & Durbin, 2010), Velvet (Zerbino & Birney, 2008) CLC (CLC Bio), and A5 (Tritt et al., 2012). Good sources for overviews of genome assemblers and the assembly process include the GAGE project (Salzberg et al., 2012), the GAGE-B project (Magoc et al., 2013), and the Assemblathon Project (Earl et al., 2011).

In this workflow, we recommend use of the open source A5 assembly pipeline which automates all of the steps described above with a single command (Tritt et al., 2012). A5 is designed to work with raw, demultiplexed Illumina data and a recent version (A5-miseq) has been optimized for longer reads from the MiSeq (Coil, Jospin & Darling, 2014). Input files should have the .fastq extension. See (http://en.wikipedia.org/wiki/FASTQ_format) for a description of the fastq format. You will need one of the two following (per genome): (1) a single .fastq file that contains both forward and reverse reads, or (2) two .fastq files, one with forward reads and one with the corresponding reverse reads. These .fastq files can optionally be gzip compressed (as indicated by the .gz file name extension). You may need assistance from your sequencing center in locating and accessing these files.

Download/Install A5 from http://sourceforge.net/projects/ngopt/.

Follow the (expert) instructions located http://sourceforge.net/projects/ngopt/files/?source=navbar

or

Follow a video made by David Coil https://www.youtube.com/watch?v=Ad6HJevC5U8

or

Follow these instructions:

After downloading and unzipping the program, change the name of the folder to a5_pipeline and move it from your downloads folder to your Applications folder. Then, create a new folder which will contain the files generated by the pipeline on your Desktop. By the way, there’s nothing special about having your file on the Desktop, it’s just there to simplify our instructions. We will refer to this folder as “a5_output”, but you should use a more informative name.

Running A5-miseq

Open a Terminal window and navigate to a5_output. A5-miseq will write all of the assembly output files to the same folder from which you run the program. In this example, the newly created folder is on the Desktop and named a5_output so the syntax for navigating to the folder in a Terminal window is

cd Desktop/a5_output/

Now that you are in the folder where you want your genome assembly to appear, you are ready to run the program. First, type or copy/paste (don’t hit return yet, and don’t copy a carriage return!):

/Applications/a5_pipeline/bin/a5_pipeline.pl

Then, drag and drop in the input file(s) into the same Terminal window (or type the path to them if you know it). Finally, type a name that will be included in all of all of your output files. So, your command line should look like this:

/Applications/a5_pipeline/bin/a5_pipeline.pl <SequenceFile1.fastq> <SequenceFile2.fastq> <MyGenomeName>

The program may take a few hours to run. Once it is completed, the terminal will display “Final assembly in MyGenome.final.scaffolds.fasta”. The complete assembly will be located in the a5_output folder.

Among the numerous files generated by A5, two of particular importance are the “MyGenome.contigs.fasta” and “MyGenome.final.scaffolds.fasta” which contain the contigs and scaffolds, respectively.

In addition, A5-miseq generates a file containing information about the quality of the assembly called “MyGenome.assembly_stats.csv” (see ‘Interpretation of A5-miseq stats’ for interpretation).

Assembly validation

There are three components to genome assembly validation. The first is the overall “quality” of the assembly, assessed by examining the stats provided by A5-miseq (discussed below). The second is verification that the organism sequenced is the organism of interest, simply by checking the assembled 16S rDNA sequence using a BLAST search (see ‘Organism Identification Using 16S rRNA Gene Sequence’ above). The third is “completeness”, which is difficult to measure without a closely-related reference genome. Nevertheless, we can get an idea of how complete the genome is by looking for highly conserved “housekeeping” genes that are found in almost every bacterial genome using a program called PhyloSift (Darling et al., 2014) (see ‘Assessing completeness with phylosift’).

Interpretation of A5-miseq stats

Open “MyGenome.assembly_stats.csv” in Excel. The first two numbers, shown in columns 2 and 3, are the number of contigs and scaffolds. Defining a “good” or “bad” assembly starts here. A finished assembly would consist of a single contig with no unresolved nucleotides but that is extremely unlikely to result from short read data. At the other extreme, we would consider a bacterial assembly in 1,000 contigs to be very fragmented. In our experience, acceptable bacterial assemblies using Illumina PE300 data, assembled with A5, tend to range from 10–200 contigs. It is also worth noting that unless studying genomic organization, the number of contigs is less important than the gene content recovered by the assembly which is typically >99% using A5-miseq (Coil, Jospin & Darling, 2014).

“Genome Size” and “Longest Scaffold” are represented in base pairs. While genome size can vary within taxa, this can be a second useful sanity check for the assembly. When expecting a 5MB genome based on other sequenced isolates from the same genus, if the assembled genome size is 2 MB or 10 MB, a red flag should be raised. “N50” represents the contig size at which at least 50% of the assembly is contained in contigs of that size or larger. This metric, combined with the number of contigs is the most common measure of assembly quality; larger is better. An N50 of 5,000bp would be quite poor, meaning that half of the entire assembly is in contigs smaller than 5,000bp. On the other hand an N50 of 1,000,000bp is considered very good for bacterial genomes sequenced with Illumina technology.

The number of raw reads/raw nucleotides “Raw reads”/“Raw nt” and error-corrected reads/nucleotides “EC Reads”/“EC nt” counts are useful for seeing what percentage of the data have been discarded. A very large difference between these numbers (“% reads passing EC”/“% nt passing EC”) would indicate either poor quality sequence data or significant adapter contamination. Adapter contamination rates can be high when the insert size is too small or if there were problems during library preparation. Poor quality sequence data can result from loading the libraries at a molar concentration that was too high for the instrument, from mechanical issues preventing focus of the sequencing instrument’s cameras, or from use of a compromised batch of sequencing reagents. Resolution of these issues would entail a discussion with your sequencing provider.

A5-miseq reports three depth of coverage statistics which can be used to assess whether sufficient data have been collected for genome assembly. First is the “Raw cov” which is simply the total number of base pairs of sequence data, divided by the assembly size. This gives an estimate of the average number of reads covering each base in the assembly. The actual number of reads at each site can and will vary substantially from the average. The second statistic is the “Median cov” which gives the median depth of coverage among all sites in the assembly. That is, 50% of sites will have greater coverage and 50% will have less than this value. “10th percentile cov” indicates a coverage level below which only 10% of sites in the assembly fall. For Illumina data, the ideal median coverage will lie between ∼20 × and 100 ×. If you have much less than 20 × median coverage, the quality of individual base calls may be compromised. Ideally, the 10th percentile coverage will be higher than 10, for similar reasons.

A separate metric of the base call quality is also reported by A5-miseq as “bases ≥ Q40”. Following assembly, A5-miseq realigns the reads to the assembled sequence and estimates the accuracy of the nucleotide called at each site in the assembly. These accuracies are provided as PHRED quality scores (Green, 2009), which represent log-scaled probabilities of accuracy. For example a PHRED score of 20 (Q20) indicates a 99% chance of the correct base, while Q30 and Q40 indicate 99.9% and 99.99% probabilities of the correct base being called. A5-miseq reports the number of assembly bases called with at least Q40.

Verification of 16S sequence

Follow the steps described in ‘Obtain the full-length 16S sequence from the assembly’, then ‘Organism Identification Using 16S rRNA Gene Sequence’, to obtain the 16S rDNA sequence from the assembly and verify that what you sequenced is what you were expecting.

Assessing completeness with phylosift

PhyloSift: Navigate to http://phylosift.wordpress.com.

Download and unzip the latest version of Phylosift.

In the terminal, navigate to the directory containing the unzipped Phylosift.

Run

./phylosift search <contig_file_name>

For example:

./phylosift search /Users/microBEnet/Desktop/Data-Genomes/Pantoea_Tatumella/tatumella/tatumella.contigs.fasta

Note: The first time you run PhyloSift it has to download a marker gene database so it may take a few minutes.

From the PhyloSift directory Move to the “PS_temp” directory.

Within this directory, Phylosift has created a directory with the same name as the input file. Move (cd) to this new directory, and then move to “blastDir”.

Open the marker_summary.txt file in the blastDir directory.

less marker_summary.txt

The DNGNGWU0001-00040 markers represent 37 highly conserved bacterial genes, if one is missing it won’t show up as a zero, it is necessary to manually verify the list. Most of the genes should only appear once. An occasional 2 is fine, but if all/a majority of the genes appear twice or even three times you have most likely sequenced multiple bacteria together. Additionally, check to make sure there is no 18S RNA sequence (at the top of the list) to ensure your sample has not been contaminated with a eukaryote (e.g., yeast).

Important Note: Markers 4, 8 and 38 are no longer included in the Phylosift analysis so do not be concerned if they are not listed. Conversely, Marker 13 is sometimes present in multiple copies and this is not a cause for concern.

Annotation

Options

Genome annotation is the process of identifying and characterizing various features of the genome such as location of genes, and putative functions for those genes. Note that we are not describing a genome “analysis” here. While genome annotation marks the final step in our workflow, it is just the beginning of a thorough genome analysis. We recommend performing this step as the bare-minimum required to include a very basic description of the genomic content for a genome publication.

There are a number of different pipelines available for the annotation of bacterial genomes. These include Prokka (Seemann, 2014), IMG (Markowitz et al., 2014), RAST (Overbeek et al., 2014), GLIMMER (Delcher et al., 2007), PGAP (Angiuoli et al., 2008) and others.

Each of these pipelines has advantages and disadvantages, and each will give slightly different results. Here we recommend RAST since it is web-based, easy to use, returns results within hours, and provides a convenient toolbox for analyzing the results. However, RAST annotations are very difficult to submit to NCBI so we recommend allowing NCBI to re-annotate the genome with PGAP upon submission. Also, we recommend reporting the annotation results from the PGAP annotations in the genome publication (for consistency).

RAST annotation

Navigate to http://rast.nmpdr.org/ and register a new account. Once you have created an account, log in. Hover over the “Your Jobs” tab at the top of the page and click on “Upload New Job”. In order to proceed you must specify a domain, a genus, a species, and the genetic code (usually “11”). Click “Finish the Upload”.

The annotation will take some time, ranging from 2 h to a few days, depending on server load. RAST will email you when it is complete. Once the annotation is complete, use their SEED Viewer to explore the annotation and metabolic pathways of the organism. From the RAST results, you can obtain information such as the presence or absence of a particular gene/pathway and you can compare the annotation to other genomes in their database.

Obtain the full-length 16S sequence from the assembly

(Skip this step if you are building the tree using the 16S rDNA sequence from Sanger sequencing).

1. Go to RAST and sign in.

2. On the “Jobs Overview” page, click on “view vetails” (under annotation progress) for the microbe you are working with.

3. Click on “Browse annotated genome in SEED viewer” (At the top of the page).

4. Click on “Browse through the features of [organism name]”.

5. In the Function column, search for “ssurna” or “SSU rRNA” (if it doesn’t work at first then refresh the page).

6. Find the ssuRNA that is 1400–1800bp in length (often Illumina assemblies also have fragments of 16S rDNA sequence that are only a few hundred base pairs long).

7. In the Feature ID column, click on the link for the sequence with the correct length.

8. Click on the “Sequences” tab (around the middle of the page).

9. Click on “Show Fasta”.

10. Click on “Download Sequences” and save as a fasta file. Rename the file to something informative.

11. Double check the identity of the sequence at BLAST: http://blast.ncbi.nlm.nih.gov/Blast.cgi?PROGRAM=blastn&PAGE_TYPE=BlastSearch&LINK_LOC=blasthome.

Data Submission

This section describes how to submit contigs and scaffolds (if applicable) as a Whole Genome Shotgun (WGS) submission to Genbank. We also recommend allowing NCBI to annotate the genome, since submitting RAST annotations to Genbank can be prohibitively complicated. The genomes are automatically shared with the DNA Data Bank of Japan (DDBJ) and the European Molecular Biology Laboratory (EBML). In addition, genomes from Genbank are automatically pulled into the Integrated Microbial Genomes (IMG) database hosted at the Joint Genome Institute (JGI), and are annotated there as well. This section also describes how to submit the raw reads, in this case we use the European Nucleotide Archive (ENA) for ease of use but the reads will be automatically incorporated into the Short Read Archive (SRA) at NCBI as well.

Before going any further you must decide if you are submitting contigs or scaffolds. Because recent versions of A5 have very good contig generation, often scaffolding doesn’t prove much additional information. For example a genome with 35 contigs in 30 scaffolds should probably be submitted as contigs only. Submitting scaffolds is significantly more complicated than submitting contigs, instructions for both are given below.

Submitting contigs only

Use this section if submitting only contigs, presumably in .fasta format.

Navigate to http://www.ncbi.nlm.nih.gov. Create an account and/or login. Then, create a BioProject at NCBI by navigating to https://submit.ncbi.nlm.nih.gov/subs/bioproject/and clicking on “New submission”. Fill in the personal information for the submitter.

Below, in italics, are the responses that we typically give for a genome sequencing project.

Project type

• Project data type-genome sequencing

• Sample scope-monoisolate

• Material-genome

• Capture-whole

• Methodology-sequencing

• Objective-assembly

Target

• Organism Name

• If you have other information feel free to add it

General info

• We recommend choosing Release immediately following curation

• Project Title

• Public Description

• Relevance-Environmental

• Biosample-blank

• Publications-blank

Once the project is submitted, refresh the page and copy down the Bioproject ID (it starts with “PRJNA”).

Create a Whole Genome Shotgun (WGS) submission

Navigate to https://submit.ncbi.nlm.nih.gov/subs/wgs/.

Click on the “New Submission” button at the top, fill in your information, and click “Continue”.

General Info

• BioProject-Yes, add the BioProject identification sequence (from the BioProject submission, starts with PRJNA).

• Biosample-No.

• Release date-Optional but we recommend Release immediately following curation.

Do not check the box stating, “Genome assembly structured comment is in the contig .sq file”.

• Assembly Method-Choose other, fill in the blank with A5 Assembly Pipeline (version can be found in the asssembly_stats.csv file).

• Version or date program was run—a5-miseq-macOS-20140521.

• Assembly name—give your assembly an appropriate name.

• Genome coverage—this is provided in the output from A5.

• Sequencing technology – Illumina (Miseq or HiSeq).

• Is this the full representation of the genome? Yes.

• Is this the final version? Yes.

• Do you intend to annotate this version? No.

• Is it a part of a multiisolate project? No.

• Is it a de novo assembly? Yes.

• Is it an update of existing submission? For most projects the answer to this will be No.

• BioSample Type: Microbe.

BioSample attributes

• Sample Name

• Organism

• Strain

• Collection date

• Geographic location

• Isolation source

• Files

• Select We have files for traditional split contigs OR gapped sequences.

• Select _ “FASTA”, upload the files

• Select “No” for the question about scaffolds

• “Is any sequence a complete chromosome?” No

• “Does any sequence belong to a plasmid” No

-Check the box below to annotate this prokaryotic genome in the NCBI prokaryotic annotation pipeline before being released. This will allow NCBI to use their PGAP pipeline to annotate the genome, and this annotation will be automatically attached to the project.

Click “Submit” and you’re done! You will receive a series of e-mails from NCBI confirming your submission and notifying you of any problems. Once the submission is pre-processed you’ll get an accession number. Note however that the data will not be released until final processing. The accession number is not acceptable for publication until after the final release of the data.

Potential problems with data submission:

Sometimes contigs that are submitted belong to contaminating organisms, or to the PhiX that is often used in sequencing. If this is the case, you will receive an email from NCBI telling you which contigs to remove. It’s important to note that after removing contigs, you need to rename all of your remaining contigs so as to not be missing numbers in the sequence. Below is a simple command that renumbers the contigs in the cleaned file (the original file with the contaminated contigs removed) and saves them to a new file (test.fa is the name of your cleaned file and test2.fa is the name you want the renumbered file to have):

cat test.fa | awk {print (NR%2==1) ? ">contigs_" ++i : $0} > test2.fa

Submitting scaffolds

Only use this section if you are submitting scaffolds, in most cases assembly with A5 will render this step unnecessary. Many of the steps are the same as for submitting contigs, only the differences are shown here.

Before submitting your scaffolded genome, you will need to have available 4–5 files which are listed below.

File types used in data submission:

• AGP file (.agp). This is a file required by NCBI to describe scaffolding.

• FASTA file (.fasta). This is the standard file type for sequence data, produced in this case by A5-miseq.

• FSA file (.fsa). Same as a FASTA file but with a different extension.

• SQN file (.sqn). The file type for sequence data required by NCBI.

• SBT file (.sbt). This is a template file type used by NCBI.

FASTA2AGP First, create the .agp file. In the terminal, navigate to the directory containing your scaffolds file. Run the fasta2agp.pl script included with A5 on the scaffold file output by the A5 assembly “my_scaffolds.fasta”.

Syntax is:

perl fasta2agp.pl < my_scaffolds.fasta> > <my_scaffolds.agp>

e.g.,

perl /Users/Madison/Desktop/a5_miseq_macOS_20140113/bin/fasta2agp.pl /Users/Madison/Desktop/a5_miseq_macOS_20140113/example/ phiX.a5.final.scaffolds.fasta > phiX.a5.scaffolds.agp

If this runs successfully then you should see both the .fsa and .agp files in your current directory.

Important Note: NCBI considers a gap of less than 10 nucleotides to be “missing information” in a contig, not a gap between contigs (whereas A5 has no minimum gap size). Therefore, NCBI requires that contigs separated by less than 10 nucleotides be merged. This script performs that merging, meaning that the number of contigs in the .fsa file may be less than in your input file. Therefore, we recommend counting the contigs in the .fsa file:

To count them in the terminal use the syntax

grep -c {\textquotedblleft}>{\textquotedblright} name_of_your_.fsa_file

Important note: If after running the fasta2agp.pl script and counting the contigs you have the same number of contigs as starting scaffolds, then you submit only the contigs as described in ‘Submitting contigs only’.

Create a SBT template Create a SBT template file at NCBI http://www.ncbi.nlm.nih.gov/WebSub/template.cgi. The BioProject # is the Bioproject ID starting with “PRJNA” which you received above. BioSample can be left blank.

When you click “Create the template”, it will automatically download to your computer as template.sbt. We recommend immediately renaming the file to the appropriate project.

Tbl2asn Download the tbl2asn program from ftp://ftp.ncbi.nih.gov/toolbox/ncbi_tools/converters/by_program/tbl2asn/.

If you are using Safari, a window will pop up asking for login information, just choose guest and unzip the version of the program that is compatible with your operating system. Other browsers will take you to a page with a lot of tbl2asn programs, download the one compatible with your operating system.

After downloading the desired command-line program, double click to uncompress the archive and rename the resulting file to tbl2asn. Now change the file permissions of the file (in the terminal) since transfer by FTP resets the permissions.

Syntax is:

chmod 755 tbl2asn

Once you have changed the permissions, create a new directory and place tbl2asn along with the .sbt file and .fsa files into the folder.

Run the tbl2asn program using the following syntax. You will need to fill out the organism name, strain, location, collection date, and isolation source specific to your own project.

path_to_program/tbl2asn -p path_to_files -t template_file_name -M n -Z discrep -j "[organism=X] [strain=X] [country=X: city, state abbreviation] [collection_date=X] [isolation-source=X] [gcode=11]"

Following the -p is the path to the directory containing the .fsa file, following the -t is the path to and name of the .sbt template file

Sample syntax

Desktop/ncbi/tbl2asn -p ~/Desktop/ncbi -t ~/Desktop/ncbi/template-1.sbt -M n -Z discrep{\textendash}j "[organism=Ruthia magnifica str. UCD-CM] [strain=UCD-CM] [country=USA: Davis, CA][collection_date=2002] [isolation-source=Calyptogena magnifica tissue][gcode=11]"

The program will output the necessary files into the directory you created earlier

(ensure no errors were generated by opening the errorsummary.val file and making sure it is blank, or listing the directory contents ($ls –lh) to ensure it has zero bytes).

Once these files are created, submission is similar to that for contigs. However, you will have to specify that you are using scaffolds and to upload the .agp file in addition to the .sqn file.

Submitting Raw Reads to ENA/SRA

We recommend using Safari or Firefox for this step, in our hands Chrome can have issues with the Java requirements for uploading files.

Go to https://www.ebi.ac.uk/ena/about/sra_submissions and create an account.

Successful creation of an account should take you to the “Welcome to ENA’s Sequence Read Archive (SRA) Webin submission system” screen.

Click on “New Submission” tab.

Select “Submit sequence reads and experiments”.

Click on “Data Upload Instructions” towards bottom of page.

This takes you to a variety of options for uploading files depending on your preference and operating system. We use the Webin Data Uploader. Click on the link which will download a .jlnp file. Open and run this file. Depending on your system you may have to download and install a new version of Java. On some systems you may have to right-click the .jlnp file and open with “Java Web Start”.

Login using your email address and password.

In the WebinDataUploader, in the blank area to the right of the Local Upload directory, navigate to the directory on your computer containing the reads (using the path as you would in the terminal).

Select the file(s) containing the reads and click “Upload”.

(Note that paired-end data is required to be in two separate .fastq files. If your data came as one interleaved file, then the separated .fastq files can be found in the directory where the A5 assembly was performed as [project name].raw1_p1.fastq.gz and [project name].raw1_p2.fastq.gz)

Note that the only acceptable file types for submission are gzip (.gz) and bzip (.bz2). To gzip files in the Terminal use the following syntax:

gzip [filename]

After completion, return to EMBL (the “New Submission” tab of the SRA Webin submission system) and select the “Next” button. During this process, refreshing the page or navigating away from the page will reset the form and the information will be lost.

Click “Create a New Study”. Fill in descriptions of the project and proceed to next tab. Select the appropriate metadata format, or in most cases the ENL default sample checklist at the bottom. Note that the default release date is three months from the current date, change this if the data should be released sooner.

You should now be at the “Sample” page. Required fields are listed on the right and optional additional fields can be selected from the options on the right. Fill out the appropriate fields and click on “Next”.

Note: If you are submitting data for an organism that doesn’t have a Taxon ID (“Tax ID”) then you need to e-mail ENA to receive one (datasubs@ebi.ac.uk). If you have already submitted the genome to NCBI then you can retrieve the Taxon ID from your BioProject page there. On the ENA page, you will be able to search for the Taxon ID and find your organism under the “Organism Details” tab but you won’t be able to find it using the name of the organism.

On the “Sample” page, click the “+ Add” button under sample group details. Fill in the unique name under basic details, add the Tax ID if it wasn’t added previously, and click “Next”. On the “Run” page, select the appropriate data type. Fill in the required fields (they change with data type).

Note: “Insert size” cannot be a range, only a number. With our 600–900bp libraries, we enter 750 here.

Click “Submit” and confirm submission. You will immediately receive a confirmation email but it takes some time before the information is actually live at the ENL links.

Publication

While submitting reads and an assembly to NCBI/ENA as described above makes the data accessible to the world, a little more effort will put that data into a more useful context. There are two general types of publications that describe microbial genomes. “Data papers” focus largely or entirely on describing the generation of the sequence, assembly, and (sometimes) annotation. “Analysis papers” dive much deeper into the genome and what the genome may reveal about the biology, evolution and ecology of the organism and its relatives. The latter (analysis papers) is outside the scope of this workflow, but a brief discussion of options for the former (data papers) is warranted. Several options exist for relatively short publications that simply describe the generation of a genome sequence. Note that all of these publications require accession numbers for the data, as described in the section on data submission above.

Our lab often publishes genome sequence “data papers” in “Genome Announcements”, a open-access journal published by the American Society for Microbiology (ASM). These papers are no more than 500 words in length, no figures are allowed, and they are not formally peer-reviewed (they do go through review by an editor). This format allows information about the generation of a genome sequence to enter the publication record with a minimum of effort.

“Marine Genomics” is an option for marine-associated microbial genome papers (“Genomics/Technical Resources”). The peer-reviewed journal is not open-access (but has that option), allows for up to 3 figures and 1,000 words but does require deposition of the strain in a culture collection.

“Standards in Genomic Sciences (SIGS)” is a peer-reviewed open-access publication with a “Short Genome Report” option. These reports adhere to a standardized format and are significantly longer than those of the previously listed options.

Another choice is the closed-access “FEMS Microbiology Letters” (which has an open-access option). This journal has a non peer-reviewed “Genome Announcement” article type which does allow figures.

For microbes associated with the gut, the open-access “Gut Pathogens” has a “Genome Announcement” article type. These articles are peer-reviewed and limited to two figures and 20 references.

Discussion

In an effort to demystify the process of microbial genome sequencing and de novo assembly, we have designed a workflow that would allow a small lab, one operating without a specialized technician or bioinformatician, to take a sample from swab to genome publication. There are many options for sequencing, assembling, and annotating microbial genomes. This workflow is only one path through the numerous choices that could be made in a genome sequencing project. All of the scripts and programs for this workflow are open-source and available online for free to ensure that individual researchers and small groups are able to access and utilize the tools necessary to complete the workflow.

Sequencing, sharing, and publishing a genome sequence can certainly be considered as an important process in its own right. Once a genome is shared, other people can use that genome for various purposes. However, just because one can stop after publishing and releasing a genome sequence that does not mean one should ignore what else one can do with the data. A genome sequence is also a starting point for many computational and laboratory analyses that can provide insight into evolution, ecology, physiology, biochemistry, metabolism, and more. Such analyses are beyond the scope of this workflow and paper but that should not be taken as implying they are not interesting, useful or important.

The authors would like to thank the many people who contributed to this workflow by field-testing various sections; Makayla Betts, Camilla Dayrit, Andrew Stump, Muntaha Samad, Henna Hundal, Cassie Ettinger, and Hannah Holland-Moritz. Additionally the authors would like to thank Authorea for technical assistance with their article platform.

Additional Information and Declarations

Competing Interests

Author Contributions

Data Deposition

The authors declare that they have no competing interests beyond the fact that this workflow recommends using software that was created by some of the authors. In addition, Jonathan Eisen is an Academic Editor for PeerJ.

Madison I. Dunitz, Jenna M. Lang and David A. Coil conceived and designed the experiments, performed the experiments, analyzed the data, contributed reagents/materials/analysis tools, wrote the paper, prepared figures and/or tables, reviewed drafts of the paper.

Guillaume Jospin and Aaron E. Darling contributed reagents/materials/analysis tools, reviewed drafts of the paper.

Jonathan A. Eisen wrote the paper, reviewed drafts of the paper.

The following information was supplied regarding the deposition of related data:

Associated data is on Figshare:

http://dx.doi.org/10.6084/m9.figshare.1064368

http://dx.doi.org/10.6084/m9.figshare.1086285.

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
