# Peer review of "Swabs to genomes: a comprehensive workflow"

_PeerJ, doi:10.7717/peerj.960_

## Round 0.1 · original submission · Minor Revisions

Please go through all comments and improvements suggested by the reviewers and improve this manuscript, before we can make final decision.

Reviewer 1 ·

Basic reporting

This is a software paper.

Experimental design

.

Validity of the findings

.

Additional comments

1. The contents in the section "General Notes on bioinformatics", which are about Linux, are not conform to the title.
2. Figure 14 should be a table format.
3. Please remove the white edge of Figure 3.
4. Figure 6 is not cited in this paper, futhermore, BLAST is a basic tool for bioinformaticians, I think that this figure is not necessary.
5. Figure 11 is not cited in the paper as well.

Reviewer 2 ·

Basic reporting

- page 3: wrong Figure 1 title: "Figure 1: Figure 1: Overview of the workow".
- line 139: paragraph sliding
- line 161: missing dot after the word "Branch"
- line 167: missing dot after the word "Node"
- line 169: "are" is unnecessary
- line 258: wrong separation (line is overgrown)
- line 359: wrong link. The right: https://www.python.org/downloads/
- line 387: wrong Figure reference
- line 400: wrong Figure reference
- line 432: in Ubuntu: "sudo apt-get install muscle" is also worked
- page 21. Figure 9. There is no "Project Name" field at the "Quick Search" page: https://gold.jgi-psf.org/simplesrch
- line 739 missing section number. Probably X=9.1.5
- line 1018: wrong citation. Rightly: [29]
- line 1341: missing url. Probably this: http://figshare.com/articles/From_Swab_to_Publication_Sample_Data_Tatumella_/1064368).
- line 1387: missing url. Probably this: (http://figshare.com/articles/Miscellaneous_Scripts_for_Workflow/1086285)

Experimental design

No Comments.

Validity of the findings

No Comments.

·

Basic reporting

This manuscript from Dunitz et al. covers a lot of ground, as it takes novice researchers from sample isolation to genome sequencing and phylogenetic classification.

In fifth grade I had to write directions on how to make a PB&J sandwich. My directions were 10 pages long and still weren't detailed enough. What I learned from this (at first glance) simple exercise was that, no matter how easy a task seems, it is incredibly difficult to write step by step instructions that all can follow. In writing workflow papers, especially ones that cover so much ground, the authors are inevitably going to have to sacrifice nuance for clarity and descriptiveness for brevity. After reading this manuscript through a couple of times, while there are certainly places where more description could be possible, the authors do a pretty good overall job at capturing the spirit of the analyses and providing a workflow that advanced high school classes could theoretically use. There are a couple of places that more depth is warranted (see below), but overall they do a pretty good job balancing thoroughness and readability.

All that being said, I think it would benefit the manuscript greatly to set up a virtual machine on iPlant (www.iplantcollaborative.org) that contains sample data sets and is set up to run most of the programs in this workflow. This virtual machine would be freely accessible to all (so long as iPlant remains accessible to all) and would provide a means to run the workflow without having to install software, get permissions, etc....Moreover, versions of this software would be frozen on this virtual machine so that anyone looking to repeat these analyses would not have to worry about changes in versions. It's a bit of work to set up one of these virtual machines, but they will be around forever and can be accessed with the click of a link. It just seems to me like it would be good to set up a one stop interface where those who were interested could forever have preprogrammed access to all the programs and analyses described in this manuscript. It's a great resource and seems like a perfect fit for this kind of workflow.

Experimental design

This section isn't quite applicable to this manuscript, but all of the described analyses and programs make logical sense. Following these directions would certainly give a bioinformatically novice user a pretty straightforward path to genome preparation and assembly.

Validity of the findings

Again, this section isn't quite applicable to this manuscript, but all of the described analyses and programs seem like they will work. I will admit to not checking every single link that the authors reference, but from what I've seen this is as good an introductory description as you can get to bacterial sampling and genome analysis.

Additional comments

The authors have a readable style of writing, but throughout I thought there were some phrases that would be better left out:

Abstract: "has become almost trivial" I would change this wording simply because "almost trivial" just reads a bit off to me in this context (especially because having done these analyses, they are never trivial".

Line 5: "and difficulty" I don't think you need these two words. IMO it's the drop in cost that is the main driver, and the level of difficulty hasn't changed, it's just been redistributed to bioinformatics.

Line 27: "relatively cheap sequencing" better as "cost efficient sequencing" or something slightly different. The words relatively cheap read too colloquial to me here.

LIne 30: "create a large activation energy" again...reads too colloquial to me.

Line 106: "It is customary to offer a small favor or gift" Please leave this line out. I understand the sentiment, but it's really weird to read in a manuscript and hopefully folks have enough humility to be thankful for the help.

Line 128: "Will often result in the isolation of pathogens" better as "can preferentially isolate human pathogens"

Line 142: Put in a temp for room temperature (given how detailed other parts of the manuscript are"

Line 152: Which online tutorial?

Line 152: "or this paper by Baldouf" better as "or Baldouf [5]."

LIne 178: It strikes me that if you are going to mention monophyletic clades, that a definition of polyphyletic for comparison sake is warranted

Line 180: "going back in time" is a bit of an unclear statement for the intended audience.

Line 183: "measure how much a particular part of a phylogenetic tree" better as "measure how well a node is supported"?

LIne 191: "sterile swab"...how can you obtain or ensure that the swab is sterile?

Line 193: "for 1-3 days" better as "until colonies of interest appear"

Line 204: "can be easily found online" better as "can be found online"

LIne 224: delete "originally developed by Fred Sanger and now"

LIne 226: "needs DNA" better as "requires DNA"

Section 6.3: You should describe the entire PCR program (annealing time and extension times, number of cycles, etc..."

Line 266: You should elaborate on "all controls behaved as expected"

Line 281: what about mentioning science exchange (www.scienceexchange.com) as a way to shop around for sequencing centers and compare prices

Line 322: seems like you need quotation marks around "upload the data without well mapping" button

Line 360: please reword "ready to go"

Line 463: "fancy" better as "complex"

Line 502-514: What about the possibility of getting human contamination from outside the sample? What would that look like? Seems like an important thing to mention given who would be using this workflow.

Line 564: what about mentioning the recent preprint showing 8$ library prep from the Baym et al? http://biorxiv.org/content/early/2015/01/16/013771

Line 708: make sure to mention to not copy/paste the carriage return either

·

Basic reporting

No Comments

Experimental design

No Comments

Validity of the findings

No Comments

Additional comments

This is a very useful protocol that will come in very handy for a large number of people. I look forward to having it available for undergraduate teaching in my own department. However it could benefit from some reorganization to make the workflow more clear. Because this is a protocol, side steps, optional steps, and parallel steps should be clearly designated. Also, the hierarchy of the organization system should be preserved across techniques.

Key suggestions:
1. Rooting the entire organizational structure around an improved workflow graphic would probably be the best place to start. I will make an attempt to include my own hasty sketch of this to make my comments more easily interpretable (See attached pdf. If nothing comes through, I'll provide upon request via email). The major sections of the workflow should correspond to major sections in the text.

2. Sections 2-4 might better be moved to the end as an appendix.

3. DNA extraction can be used in parallel for 16S analysis and genome sequencing, so this creates a set of two parallel processes from that point.

4. Make all the processing steps (RDP, SeqTrace, custom scripts) subsections of a larger whole (e.g., Sanger Sequence Processing)

5. Add a section on genome sequencing. Right now there is a section for Sanger sequencing, but nothing for the actual genome sequencing.

6. As a subset of assembly, add some comments about demultiplexing options, since multiplexing is discussed previously in some detail.

7. Demote section 10 to a subset of annotation

8. Break out section 11 as an analysis methodology that can be used in service of the first parallel chain (16S ID) and the second (whole genome sequencing).

9. Add a section on publication. There are several options, from genome announcements at J Bact, like you have done, to SIGS, Marine Genomics and PLOS. These have different appeals and drawbacks and make a logical conclusion to the steps herein.

10. The last paragraph of section 7.4 (lines 547-553) is a big issue and deserves to be extracted and expanded upon in the introduction or as an appendix item like the other “General Notes” items. Any considerations here that might help people select strains for sequencing that are of the most benefit to the community will add value to their efforts.

Minor comments
Section 2.2- change to “Summary of common commands and terms”
In the flow chart, “colony PCR” is used, but “direct PCR” is used in the text.
Section 5.2- include a reference for a plate making protocol.
Section 6, line 218- following “liquid culturing,” instead of “second dilution streaking”
Line 225, 257, 265, elsewhere, remove the reaction from “PCR reaction”- it’s redundant.
Line 240, “…use of a phenol and chloroform extraction…”
Line 387, 400, elsewhere, figure designations in the text are messed up
Add Figures 6-8 to the text somewhere. I didn’t see them mentioned.
Section 7.4 is really parenthetical to the regular blast-based approach mentioned earlier, since you will get many alignments to view. It might be worth adding/combining this to the processing steps.
Figures 7 and 8 need some kind of discussion in the legend about where the trees came from, and why there are being used, because as they are positioned in the text, the novice may wonder why they don’t get trees from their blast results.
Line 610- …multiplex (Section 8.10) to achieve…

---

## Round 0.2 · accepted · Accept

We are pleased to inform you that you have addressed the minor issues and the MS is acceptable.